# Tsallis Entropy and Mutability to Characterize Seismic Sequences: The Case of 2007–2014 Northern Chile Earthquakes

**DOI:** 10.3390/e25101417

**Published:** 2023-10-05

**Authors:** Denisse Pasten, Eugenio E. Vogel, Gonzalo Saravia, Antonio Posadas, Oscar Sotolongo

**Affiliations:** 1Department of Physics, Universidad de Chile, Santiago Las Palmeras 3425, Santiago 8330111, Chile; 2Department of Physics, Universidad de La Frontera, Temuco Casilla 54-D, Temuco 4780000, Chile; 3Center for the Development of Nanoscience and Nanotechnology, Universidad de Santiago de Chile, Santiago 9170022, Chile; 4Los Eucaliptus 1189, Temuco 4812537, Chile; gonzalo.saravia@gmail.com; 5Departamento de Química y Física, Universidad de Almería, 04120 Almeria, Spain; aposadas@ual.es; 6Instituto Andaluz de Geofísica, Universidad de Granada, Campus Universitario de Cartuja, 18071 Granada, Spain; 7Cátedra de Sistemas Complejos ‘‘Henri Poincaré’’, Universidad de La Habana, Habana 10400, Cuba; osotolongo@gmail.com

**Keywords:** Tsallis entropy, information theory, subduction seismicity

## Abstract

Seismic data have improved in quality and quantity over the past few decades, enabling better statistical analysis. Statistical physics has proposed new ways to deal with these data to focus the attention on specific matters. The present paper combines these two progressions to find indicators that can help in the definition of areas where seismic risk is developing. Our data comes from the IPOC catalog for 2007 to 2014. It covers the intense seismic activity near Iquique in Northern Chile during March/April 2014. Centered in these hypocenters we concentrate on the rectangle $Lat_{-22}^{-18}$ and $Lon_{-68}^{-72}$ and deepness between 5 and 70 km, where the major earthquakes originate. The analysis was performed using two complementary techniques: Tsallis entropy and mutability (dynamical entropy). Two possible forecasting indicators emerge: (1) Tsallis entropy (mutability) increases (decreases) broadly about two years before the main $M_{W}8.1$ earthquake. (2) Tsallis entropy (mutability) sharply decreases (increases) a few weeks before the $M_{W}8.1$ earthquake. The first one is about energy accumulation, and the second one is because of energy relaxation in the parallelepiped of interest. We discuss the implications of these behaviors and project them for possible future studies.

## 1. Introduction

We can approach a variety of problems in physics through statistical mechanics. Some examples include real magnetization systems [1,2], spin models [3,4], molecular interactions [5], fluids [6], space plasmas [7,8] among others. However, statistical mechanics can be also useful in more complex systems such as social interactions [9], traffic [10], wind energy [11], and earthquakes [12,13,14,15,16,17]. A common key element in this variety of applications is entropy, which directly points to the accessible states under given conditions. Magnetic systems and rocks under tension could both alter their configuration spaces making some external manifestations more probable/improbable. When we speak about probability of states we are reaching the domain of entropy, that “lives” in the configuration space. Entropy can be defined in different ways. This paper focuses on two forms: Tsallis entropy and mutability (or dynamical entropy).

In recent years, studies of entropy in earthquakes have been used to show the evolution of the seismic systems in time. These studies have relied on datasets coming from different zones of the Earth. In a quick summary, we can mention the following recent developments: (i) Shannon entropy has been found useful in identifying earthquake risk areas [18]; (ii) a study established the way both the Shannon entropy and mutability reflect the seismic activity [19]; (iii) researchers have recently tested Tsallis entropy in different seismic zones of the world [20,21].

In addition, in all the studies previously mentioned we can use the concept of natural time. This concept has proven to be very useful in the study of earthquakes. The natural time allows one to follow a time series step-by-step through a scaling of the time such as $\chi_{k}=k/N$ where *k* follows the occurrence of a seismic event in time [22,23]. In the present paper, we will use it for comparison only so a simpler form is enough: the enumeration of events.

In the present article, we revisit the same seismic area of our recent articles [19,24] with the purpose of completing the study with the following recent developments: (a) for the first time we report mutability on the sequence of magnitudes (before we investigated it on intervals); (b) for the first time we compare and discuss Tsallis entropy and mutability on the same footage: this allows us to call the later “dynamical entropy”; (c) we perform a tuning process to detect the importance of the size of the time window to analyze the dynamical process; (d) we conduct a progressive approximation to days and hours prior to the large earthquake to detect premonitory signs and we believe we can report a couple of them; (e) we conclude that the aftershock regime closed quickly in this area and energy continues to accumulate at levels similar to those before the 2014 earthquakes.

## 2. Methodology

### 2.1. Data Source

Chile is placed close and almost parallel to the border of the subduction zone between the Nazca Plate and the South American Plate. This is a source of seismicity in a wide range of magnitudes along different geographical conditions. In recent years, scientists have concentrated their attention on the seismicity of the northern zone of Chile. IPOC is an outcome of this effort, which is a network of institutions from Europe and South America. Its networks have measured earthquakes on the Peru-Chile coastal margin for decades. This network’s seismic data is helpful to understand the seismic dynamics in northern Chile and to identify potential risks.

Scientists are interested in the Northern zone of Chile because of its frequent earthquakes and the fact that there have been no major earthquakes in the recent past. The last historical mega-thrust earthquake in the northern zone of Chile was in 1877 [25,26,27]. A partial list of important recent earthquakes is: Antofagasta (1991) $M_{w}$8.0 [28], Tarapaca (2005) $M_{w}$7.7 [29], Tocopilla (2007) $M_{w}$7.7 [30], and Iquique (2014) $M_{w}$6.6, $M_{w}$8.1, $M_{w}$7.6 [31,32]. Each one of the previous large seisms generated a powerful chain of aftershocks. This seismicity is mainly shallow at intermediate depths (less than 80 km). The quality of the data has improved with time due to better stations, more stations, and more coordination among seismological institutions. This is the main reason to consider only a few years before 2014, up to 2007 considering the Tocopilla seism as the last previous event.

The Iquique earthquake has a complex structure that deserves a dedicated investigation. In its simplest form, it can be viewed as a triple earthquake in 2014: $M_{w}$6.6 on 03.16, $M_{w}$8.1 on 04.01 and $M_{w}$7.6 on 04.03. Each of the previous earthquakes generated aftershock activity. Even the first one (the weakest of the three) had two important aftershocks $M_{w}$6.4 on 03.17 and on 03.23, and several others over $M_{w}$6.0. Alternatively, one can consider that the $M_{w}$8.1 seism is the important one here, declaring all previous activity in the area as pre-shock and what came afterwards as aftershock activity; the $M_{w}$7.6 earthquake is absorbed within the aftershock of the larger one. However, one can also choose to consider this last seism on its own, with its aftershock regime and rupture area. This is an anticipation of the results and discussions to be represented below.

It is also important to consider some previous discussions concerning the aftershock activity in this region [31,32,33], and a zone with a low coupling [27]. Socquet et al. in 2017 [34] showed that the major shock was led by an acceleration that started aproximately eigth months before the large earthquake. Jara et al. 2017 [35] found a strong link between shallow and intermediate depth seismic activity, showing that it may have caused the Iquique earthquake. All this evidence points to the understanding of the physical process behind the occurrence of a great earthquake and it is to this understanding that this work also contributes.

### 2.2. Handling of Data

We center our attention on the large Iquique earthquake (1 April 2014) with $M_{w}$ 8.1 located at 19.589° South Latitude and 70.940° West Longitude; its depth was 19.91 km (data from IPOC catalogue [36]), its preshock activity and the aftershock activity recorded in the IOPC catalogue. The epicenter was situated 95 km Northwest of Iquique city and a tsunami alert was issued for Chile, Peru and Ecuador, which was later extended to Colombia and Panama. Viewed in this way, the major shock is preceded by an intense foreshock sequence and followed by a large $M_{w}$7.6 [37] as is shown in Figure 1.

Preparing the initial dataset comprises three steps. First, we chose the epicentral area according to the main event coordinates. We drew a “rectangle” with $Lat_{-22}^{-18}$ and $Lon_{-68}^{-72}$ based on the main event coordinates, and found 65,050 seisms in the IPOC catalogue. We considered all these seisms to calculate the Gutenberg-Richter relationship to define the threshold magnitude $M_{0}$ (we used the MAXC technique [38] because it is a simple method for the goals we pursue). The second step involved setting up a $M_{0}$2.2 that is shown in the peak of the red triangles in Figure 2.

We consider all the seisms within the parallelepiped defined by $Lat_{-22}^{-18}$ and $Lon_{-68}^{-72}$ and 200 km depth. Longitude and depth are used as coordinates to make a map of all earthquakes, regardless of latitude. Dots represent the location of seisms in Figure 3. A careful look at this figure unveils the two plates, with the subduction front defining a downward diagonal from West to East.

The third step is the right panel of Figure 3 where now depth is the only variable while seisms result in a histogram giving the abundance of seisms as a function of depth. A bimodal is clearly appreciated, where the lower component receives most of its contributions from a mixture of tectonic (Continental Plate) and intra-plate earthquakes. The large deep distribution is originated within the Nazca plate. The first group of earthquakes deserves our full attention, as those earthquakes caused the most damage and could trigger deadly tsunamis. For this reason, we set the deepness filter at 70 km, corresponding to the minimum of the distribution in the right panel of Figure 3.

Moreover, we left out the shallower first 5 km to avoid contamination coming from the mining work conducted in the area. Earthquakes from 5 to 70 km will be handled in the rest of this paper. We make this cut in the dataset to focus this analysis in the zone close to the hypocenter. The number of earthquakes left for study after previous filtering is 10,640.

Finally, Figure 4 presents the epicentral zone with the selected earthquakes.

### 2.3. Tsallis Entropy

Let start by considering an earthquake as a critical phenomenon in a complex system (fracture zone) that experiences a phase transition from a non-equilibrium state (where stresses and strain in crust lead to fault slip) to another state (where stresses and strain have become to relax); several physical models have been developed to describe their essential properties [13,15,23,39,40,41,42]. Thereby, the maximum entropy principle has widely been applied in many out-of-equilibrium systems in physics (and other sciences), providing novel insights into their macroscopic states [43]. Sotolongo-Costa and Posadas (2004) [20] introduced the fragment-asperity interaction model for earthquake dynamics (SCP model) based on the non-extensive statistical formalism; in this model, the released seismic energy is related to the size of the fragments that fill the space between fault blocks. According to the SCP model, if *N* (>*M*) is the cumulative distribution of the number of earthquakes *N* with magnitude greater than *M*, then:(1)$$\log\left(N\left(>M\right)\right)=\log\left(N\right)+\frac{2-q}{1-q}\log\left(1+a\left(q-1\right){\left(2-q\right)}^{\frac{1-q}{q-2}10^{2M}}\right).$$
where *a* is a real number expressing the proportionality between the released seismic energy and the size of the fragments, and *q* is the entropic index from Tsallis entropy. Equation (Equation 1) appropriately generalizes the Gutenberg–Richter relationship over a broad range of magnitudes [43] and exhibits an excellent fit to earthquake datasets [14,20,44,45]. In fact, the Gutenberg–Richter law can be easily deduced as [12,46]:(2)$$b=2\frac{2-q}{q-1}.$$

Moreover, *q* values obtained from different regions of the world [12] are all q≈1.5−1.7, suggesting the universality of this constant.

Recently, Posadas and Sotolongo-Costa (2023) [24] established the entropy of fragments and asperities within fault fractures (i.e., within gouge fault zones) and determine their behavior during an earthquake. Authors assume the hypothesis that prior to an earthquake, the state of the system, characterized by a range of fragment sizes and stress distribution forms many “microstates” compatible with fragment distribution; such entropy can be assumed to be (relatively) large. During an earthquake, fragments are broken, while asperities and barriers are overcome. Furthermore, fragment sizes become homogenized and this decreases the number of possible “microstates”, as such, entropy decreases. As this process is abrupt and rapid, the entropy decreases suddenly; it subsequently recovers as stress starts to re-accumulate. From a statistical mechanics perspective, the higher the number of microstates, the higher the entropy and vice versa.

Tsallis entropy (Equation (Equation 1)) for a continuous distribution p(σ) of fragments of sizes σ is given by (for simplicity we set k=1):(3)$$S=\frac{1-\int_{0}^{\infty }p^{q}\left(\sigma \right)d\sigma }{q-1},$$
subject to two restrictions:(4)$$\int_{0}^{\infty }p\left(\sigma \right)d\sigma =1$$
and
(5)$$\int_{0}^{\infty }\sigma p^{q}\left(\sigma \right)d\sigma ={\langle \langle \sigma \rangle \rangle }_{q},$$
where ${\langle \langle \sigma \rangle \rangle }_{q}$ is the mean of the distribution. Therefore, the maximum entropy principle allows us to form the following Lagrangian:(6)$$L\left(p\right)=\frac{1-\int_{0}^{\infty }p^{q}\left(\sigma \right)d\sigma }{q-1}-\alpha \int_{0}^{\infty }p\left(\sigma \right)d\sigma -\beta \int_{0}^{\infty }\sigma p^{q}\left(\sigma \right)d\sigma .$$
where α and β are the Lagrange multipliers. Imposing the Lagrangian to be extreme:(7)$$\frac{\partial L}{\partial p}=0$$
after some algebra it is possible to find that:(8)$$p\left(\sigma \right)=\frac{{\left[\frac{1-q}{q}\alpha \right]}^{\frac{1}{q-1}}}{{\left[1+\beta \sigma \left(q-1\right)\right]}^{\frac{1}{q-1}}}\;,$$
where, implicitly, a cut-off condition has been used for the denominator [47]. Finally, by substituting Equation (Equation 8) into that of non-extensive entropy (Equation (Equation 3)) and solving the integral in the numerator [21], we can obtain:(9)$$S=\frac{1-\int_{0}^{\infty }p^{q}\left(\sigma \right)d\sigma }{q-1}=\frac{1-{\left(2-q\right)}^{\frac{1}{2-q}}}{q-1}.$$

This equation allows us to find the value of the entropy for a dataset and to study its behavior as a function of the non-extensive *q* parameter; therefore, if a windowing process is carried out (i.e., choosing a certain number of earthquakes and sliding the window in time), it is possible to visualize the dynamic evolution of the seismic series in terms of non-extensive entropy. The process is as follows:

1. First, the time window *W* is determined for the calculation of entropy; in other words, the minimum number of earthquakes used to calculate *S* from Equation (Equation 9). In general, the final window size is a reasonable compromise between the required resolution and smoothing results.

2. Second, parameter *b* from the Gutenberg–Richter relationship for the chosen window *W* is determined; this can be calculated from the classical expression of Aki (1965) [48] and the subsequent correction by Utsu (1965) [49]:(10)$$b=\frac{\log(e)}{\bar{M}-\left(M_{0}-\frac{\Delta M}{2}\right)},$$
where $M_{0}$ is the threshold magnitude; ΔM is the resolution of the magnitude (usually ΔM=0.1); and $\bar{M}$ is the average value of all possible magnitudes, which is given by:(11)$$\bar{M}=\int_{M_{0}}^{\infty }Mp\left(M\right)dM.$$

The estimation of $M_{0}$ is performed, as we noted before, using the maximum curvature (MAXC) technique [38].

3. Finally, approximation according to Sarlis et al. (2010) [12] Equation (Equation 2) is used to determine *q*; then, the non-extensive entropy is computed for each time *t* following Equation (Equation 9). By convention, the time attributed to each point of the analyses is the time of the last seismic event considered in each window.

### 2.4. Mutability

Information content is valuable information leading to entropy in different ways [50,51,52]. During the last decade or so a dynamical entropy called mutability has been introduced in an empirical way to characterize information content in a data sequence [53].

To obtain the value of mutability, we first create a vector file with the sequence or time series to be recognized (Monte Carlo simulation of the magnetization of a system, magnitude of consecutive earthquakes in a given region, variations in the value of a given economical asset, and similar sequences of measurable evolving quantities). All registers have the same number of digits filling with zeroes the empty positions. The number of bytes occupied by this vector file is *w*. This file is then compressed and the compressed file occupies $w^{*}$ bytes. Then, the value of the mutability for this sequence is defined as the ratio:(12)$$\mu \left(\alpha \right)=\frac{w^{*}}{w}\;,$$
where α represents the set of parameters that characterize the system (size, temperature, etc.).

In principle any data compressor can accomplish this task and blzip2 was used previously [54]. However, data compressors are based on search for repetitive chains of characters, which can accidentally occur without a physical meaning. To cope with this inaccuracy, a data compressor based on exact matching of physically meaningful information was developed under the name “world length zipper” (wlzip for short) which will be summarized next [54].

### 2.5. Algorithm of the Data Recognizer

The compressed or recognized file is a map constructed from the original, according to an algorithm that obeys the following rules:(a)Navigate to the first register of the original file, copy it onto the compressed file as a first register followed by a space and then the digit 0 to indicate the beginning or origin of the new file.(b)Select the following register in the original file and compare it to the already stored register(s) in the compressed file.-If this register already exists then navigate to its row, leave a space and write to the right the “distance” or number of registers since it was previously found in the original file.-If the register repeats itself immediately after, place a comma and the number of consecutive repetitions.-If this register is new, then write it at a new row followed a space and then the distance to the first register.(c)Navigate to next register and repeat the procedure given in (b) until the last register in the file.

So now we have the original file with weight *w* and the compressed file with weight $w^{*}$. According to Equation (Equation 12) the mutability of the original file measured in this way is $w^{*}/w$.

We illustrate the concept of mutability and the use of wlzip by two sequences of 50 seisms each, both obtained from the filtered catalog defined above.

The first sequence, called “Before”, lists the magnitudes of the 50 seisms beginning on 1 February 2014, covering a few days before the 6.6 earthquake. Their sequential magnitudes are: 2.4, 4.0, 3.8, 4.0, 2,2, 4.6, 3.5, 2.9, 2.2, 3.0, 4.1, 3.1, 3.1, 4.3, 4.3, 3.6, 5.5, 3.5, 2.4, 3.5, 2.7, 2.5, 2.2, 2.5, 2.3, 2.5, 2.6, 3.3, 2.4, 2.9, 2.6, 2.8, 2.3, 2.2, 2.2, 2.4, 3.8, 4.1, 2.8, 2.4, 2.5, 2.2, 2.5, 2.8, 2.3, 2.5, 3.4, 4.0, 4.0, 2.9.

The second one, called “After”, gathers the sequence of magnitudes of the 50 seisms beginning with the 6.6 earthquake of March 16, 2014 and obtains the 49 following seisms. The list is the following: 6.6, 4.5, 4.8, 4.1, 3.1, 5.2, 4.8, 4.2, 3.8, 3.5 3.1, 4.8, 3.7, 3.1, 3.6, 3.3, 3.1, 4.0, 2.7, 3.5 3.2, 4.8, 2.8, 3.5, 2.4, 4.7, 3.0, 2.9, 3.2, 4.1 4.3, 4.0, 5.1, 3.3, 4.3, 3.4, 3.8, 4.7, 3.1, 3.1 2.6, 4.7, 3.4, 3.3, 3.7, 4.7, 3.0, 3.1, 3.7, 3.4.

The procedure is illustrated in Table 1. The left-hand-side is the compressed file of “Before” called “BeforeC” and the right-hand-side is the compressed file of “After”, namely, “AfterC”.

Let us begin with “Before”. We obtain its first register in the first row of “BeforeC” followed by its position 0 at the origin. The following register is 4.0 (new) and just one position from the first one. The next register is 3.8 (new), two positions from the origin. The next one is 4.0, which is already listed so we navigate to its row in BeforeC and write a 2 to the right, meaning that its new position is two rows below its previous appearance. The next register is 2.2 which is new and four positions from the origin. Similarly, we continue with the new ones 4.6, 3.5 and 2.9. However, then we find 2.2 which we found found positions before so we add a 4 to its row. We continue to the magnitude 3.0 found nine positions from the origin. Next is also a new one: 4.1, ten positions distant from the origin. The following one is 3.1, also new, but repeats itself immediately so we write its coordinate 11 then a comma and then the number of repetitions which is two. Something similar happens with magnitude 4.3 coming next. In this way we can continue applying the rules above to complete the file “Before”. We perform a similar procedure to the file “After”.

One first results is obvious: The length of “After” is larger than the one of “Before” and so it is its span of values. When the mutability is measured for these two files the result is 0.94 for “Before” and 1.00 for “After”. It is clear that the aftershock regime brings in a larger variety of magnitudes so the information content increases and so does the mutability (dynamical entropy).

Columns identified by $f_{B}$ and $f_{A}$ give the abundance or frequency of this value in the sequence. So these columns give the histograms for the sequences Before and After, respectively. However, columns marked as MapB and MapA also represent a histogram, but with an internal structure related to the dynamics of the sequence. This is the basis of the difference between the mutability and other forms of entropy where it is only the distribution of values that matters, regardless of the way the sequence was produced.

### 2.6. Tuning the Information Recognizer

The algorithm is now a powerful program that we offer for free—email ee.vogel@ufrontera.cl to download it. This allows us to distinguish and process different data according to what is appropriate for each system. The following adjustments need to be made:(i)Is this a static calculation (entire file, just once) or a dynamical calculation through time windows? Answer: it is dynamic through windows with *W* registers.(ii)Are these successive independent or overlapping windows? Answer: we use overlapping successive windows.(iii)If they overlap, what is the size of the overlap? We consider here a displacement of just one register between consecutive windows so the overlap is W−1 events.(iv)In step (b) of the algorithm described above, a numeric comparison is performed between two registers. How many digits and which digits bear the most sensitive information to perform this comparison? An estimation is possible after inspecting the data, but we let wlzip itself find the digits that lead to a better precision. The comparison is restricted to the *r* digits from position i and the following r−1 digits; this is denoted (i,r). In the examples of Table 1, all comparisons were for i = 1, r = 3 (the dot needs to be compared as well).(v)If precision is needed, wlzip has the feature of handling different numeric bases (quaternary, binary, ...) which can help to discriminate intermediate positions.

WLZIP applies to any parameter P(t) stored in a vector file and indeed it has been used to recognize phase transitions or criticality in different fields: magnetism [54,55], econophysics [56], polymer deposition on surfaces [57,58], wind energy optimization [59].

The first application of wlzip to seismology came recently using data from a Chilean catalogue measured by CSN [60] finding the variations in wlzip results years and months prior to large earthquakes [61]. Then the study was extended to four zones along the subduction trench, comparing their dynamics by means of Shannon entropy and mutability [19]. More recently, a deepness analysis of the same “rectangle” near Iquique was performed by means of Tsallis entropy [24].

## 3. Results

The earthquakes of 2014 have raised several questions concerning their dynamics. Here, we addressed their study computing the Tsallis entropy for the first time focusing on the magnitude of the quakes, complemented by the mutability on the same data. For the first time, we will compare these two entropies based on an analysis over time.

Figure 5 plots the Tsallis entropy for successive overlapping time windows defined by the last *W* seisms. (A) W = 256; (B) W = 512; (C) W = 1024; (D) W = 2048. Abscissas reflect real time in days with major ticks close to a year mark. Let us examine plot (A) where, apart from oscillations, we see a broad valley around day 800, a maximum or “swelling” around day 1600, and stronger oscillations with a sharp decrease near day 2600, which roughly coincides with the big earthquake. Then, Tsallis entropy partially recovers in an oscillatory way during the aftershock regime. In Figure 5B–D we can observe similar behaviors, except that oscillations are damped due to a larger statistics upon increasing the window spans. The larger *W* values also displaces the texture of the curves a bit to the right. The three main earthquakes that mark these complex seismic behaviors are shown by stars with the magnitudes displaced in the inset. They were vertically split since otherwise they would overlap at this scale.

It is interesting to notice how these four figures show a consistent increment in the Tsallis entropy before an abrupt decay. We believe this is a manifestation of the subduction process where the large fragments in between the Nazca plate and the South American plate prevent sliding. However, possibilities in which the dynamics can change increase with time leading to an increase in Tsallis entropy [21]. This effect begins about two years before the large earthquake which is in agreement with Socquet et al. in 2017 [34], who detected an increase in the acceleration in the displacement of the plates several months before the large earthquake.

When the fragments fracture, smaller pieces tend to fill in the interspace, thus lubricating the sliding of one plate with respect to the other. This leads to a sudden decrease in the Tsallis entropy to denote the time of the major seism and the beginning of the aftershock regime.

In addition, we should mention that prior to the large earthquakes events are mostly independent: their epicenters are at different locations, no time correlation is observed, and magnitudes are moderate. Thus, during the apparent calm period, mostly uncorrelated seisms are produced. The large $M_{w}$8.1 earthquake fractures the fragments, unleashing a variety of correlated seismic chains in little time, causing an abrupt decrease in Tsallis entropy. As the underground layers settle, the near equilibrium goes back to the situation months or years before the violent earthquake.

In Figure 6A–D we present the mutability results for the same seisms, using the same time windows of previous figure. In all cases the plots sharply increase when the moving windows reach the time of three main earthquakes; for more clarity their times are marked using particular symbols as given in the inset. The main feature of these figures is the abrupt growth of mutability during the earthquake period. Larger time windows moderate the oscillations, but the sharp peak near day 2600 prevails in all of them. A more subtle feature becomes more pronounced as the time window increases: for years before the activity of 2010, the mutability goes through a maximum then it decreases reaching a minimum. It then quickly recovers, maximizing during the great seismic activity.

Figure 5 and Figure 6 show an inverse behavior. To understand the behavior of the mutability we must remember that it is essentially based on the special kind of histogram constructed in the way shown in Table 1. Just before the main seisms, the series are comparable to the case of BeforeC in Table 1: lower mutability. During and after the large seisms, the mutability is closer to the situation represented by the column AfterC in Table 1, namely, larger mutability values. So the mutability decreases during the calm periods before an earthquake at the same time the Tsallis entropy grows. During the large seisms and their immediate aftershock activity the mutability sharply grows at the same time Tsallis entropy sharply decreases. Turning to the aftershock period, Tsallis entropy gradually increases while the mutability gradually decreases, both in oscillatory ways.

In Figure 7 we use a window of 512 seismic events and we compute the Tsallis entropy and the mutability before the large earthquake of magnitude $M_{w}$8.1, stopping short before including the main earthquake. Thus this series presents the instant picture 80, 40, 20 and 2 days before the major earthquake. Several features are of interest here. First, Tsallis entropy and mutability progress in complementary ways; this is probably due to the fact that Tsallis entropy is based on the real space, while mutability looks at the states in the configuration space. Second, a rather pronounced change in the dynamics of both Tsallis entropy and mutability is already noticeable 80 days before April 1, where no important seism has been reported. Third, this is confirmed 40 days prior the large earthquake, where both curves present a slight turn back. This picture remained frozen until 20 days before. Then, in the picture taken 2 days before, we have a pronounced change in both curves—a product of the 6.6 seism of March 16 and the subsequent aftershock activity. Fourth, this premonitory behavior is additional and shorter in time than the previously mentioned maximization (minimization) of the Tsallis entropy (mutability) around the years 2011 and 2012, namely, two years in advance.

These results show how the Tsallis entropy and the dynamic entropy could notify a change in the configuration of the seismic system days before the occurrence of a large earthquake being a contribution to the evaluation of seismic hazards in this zone. Eventually, this way of introducing the approach to the critical moment could be presented in video formto further stress the anticipation signals the system emitted before the breakdown.

Finally, Figure 8 is a remake of Figure 6 but now using the 10,640 seismic events as natural time along the abscissa axis. In this way the sequence is better appreciated. Thus, for instance, the symbols marking the three main earthquakes indicated in the inset open up, allowing to appreciate the role of each seism in each plot. The decrease in mutability before the leading earthquake is clearly manifested, especially for W = 512 and over.

More work has to be performed before claiming this could evolve into a method to diagnose seismic risk, but at the moment we leave it as a proposal or hypothesis. To delve deeper into this matter, catalogues of quality similar to IPOC are necessary, validated over many years or decades in different regions of the world. We shall attempt to conduct this with time whenever is possible.

## 4. Conclusions

The accommodation of the ground layers under the Earth’s surface is accompanied by a variation in the energy of the system which is at times released in sudden ruptures and slides with catastrophic consequences in the events known as earthquakes. However, this transit from states prior and after the seism mean also a change in the entropy of the system (“cube” formed by the rectangle on the globe and the defined deepness).

This is an out-of-equilibrium system, since external fields act on it to provoke changes. In our case, the Nazca plate is coming from the West submerging under the Continental plate. Asperities make this process a discontinuous one. For many years, large rocks can prevent the flow of the plates until they fracture. The number of possible states for the system is not constant and varies with time according to the hidden physics under the planet’s surface. However, we can have an indication of the way they change looking at the sequence of seismic data they produce.

The simultaneous application of both Tsallis entropy and mutability (dynamical entropy) to the magnitude series for the first time proved to render valuable information. Despite that magnitudes span no more than about 50 different values between 2.2 to about 7, they can recognize meaningful states on which a statistical analysis can be performed. In the past, mutability was applied to interval series [61], where meaningful values span about three orders of magnitude with intervals expressed in minutes as the time unit.

Tsallis entropy of magnitudes grows during the time the system accumulates tension and energy. Mutability on the same magnitude series decreases during that period. This behavior can be highlighted by choosing appropriate observation windows to analyse the dynamics. Thus, Tsallis entropy responds to the conditions in the real space, while mutability is driven by the accessible states in the configuration space.

We can perform the analysis of the seismic sequence either in real time or in natural time. Therefore, the combination of both pictures is very useful for a detailed understanding of the dynamical process. The most recent results within the data provided by the IPOC catalogue show that this zone recovered rather quickly to the conditions before the major earthquakes, which is a clear sign that energy accumulation resumed.

## Figures and Tables

**Figure 1 entropy-25-01417-f001:**
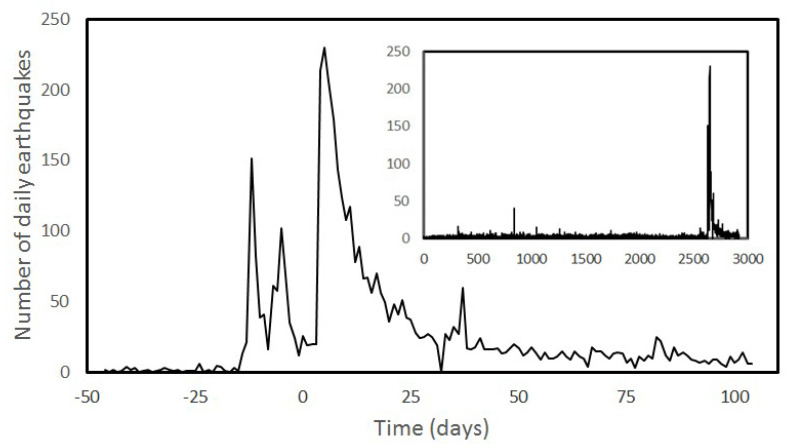
Number of daily earthquakes in the selected IPOC catalogue from years 2007 to 2014, fully displayed in the inset. On the abscissa axis in the main body, day 0 corresponds to 1 April 2014, coinciding with the $M_{w}$8.1 earthquake. It can be noticed that the pre−shock activity appears about 17 days before. The aftershock response extended months afterwards.

**Figure 2 entropy-25-01417-f002:**
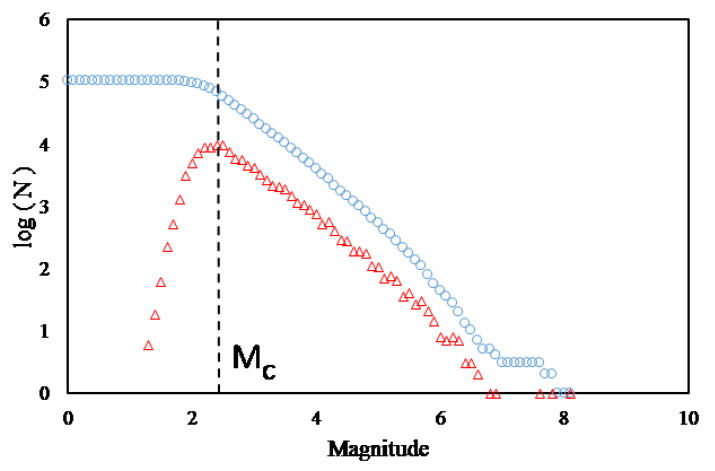
We analyzed the data from the IPOC catalog using the Gutenberg-Richter law, including earthquakes from 2007 to 2014 with epicenters within 18 °S–22 °S and 68 °W–72 °W. Circles denote the cumulative number of earthquakes; triangles denote the abundance of earthquakes for a magnitude. Based on the maximum curvature (MAXC) technique (Wiemer and Wyss, 2000), $M_{0}=$ 2.2.

**Figure 3 entropy-25-01417-f003:**
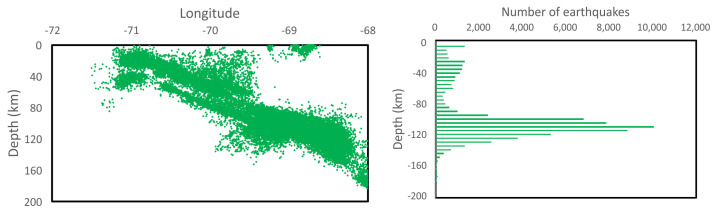
Depth distribution of earthquakes at different longitudes. Seisms at different latitudes are accumulated on this two−dimensional view. A histogram with respect to depth is presented on the right panel.

**Figure 4 entropy-25-01417-f004:**
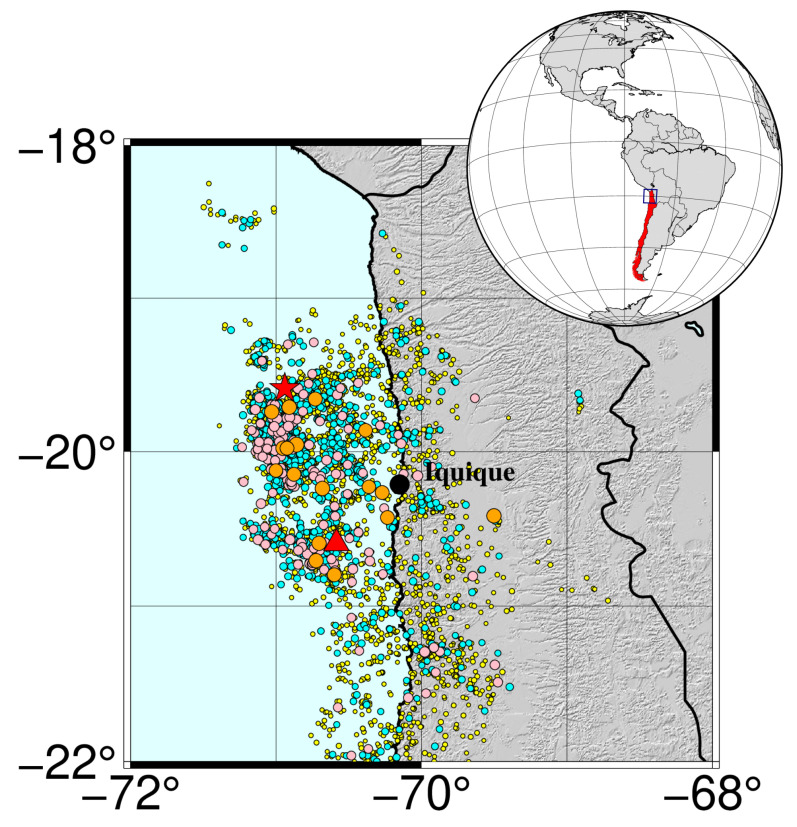
Epicentral map of the area under study: the black circle marks the city of Iquique, 95 km from the epicenter. Magnitude of the seisms are illustrated by both a proportional diameter and the color of the circles: yellow $3.0\leq M_{w}\leq 3.9$, cyan $4.0\leq M_{w}\leq 4.9$, pink $5.0\leq M_{w}\leq 5.9$, orange $6.0\leq M_{w}\leq 6.9$. The red star positions the great earthquake of Iquique with $M_{w}$8.1, while the red triangle shows the epicenter of its main aftershock with $M_{w}$7.6.

**Figure 5 entropy-25-01417-f005:**
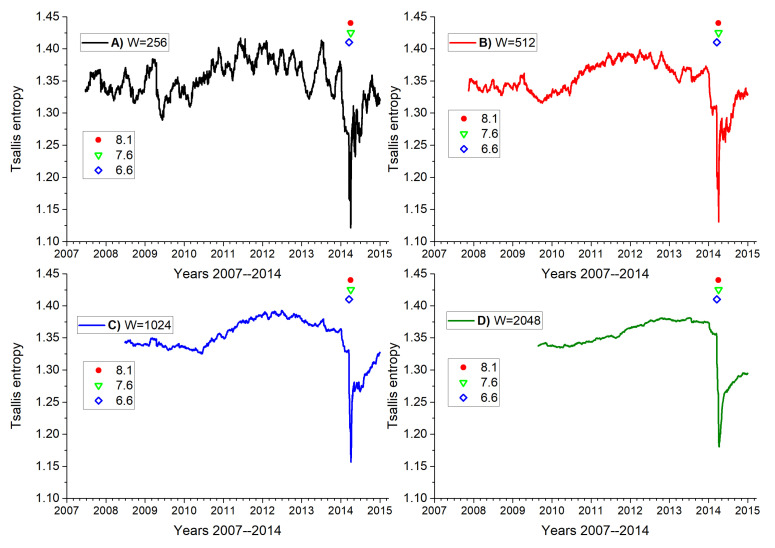
Tsallis entropy on magnitude sequence in terms of real time, using four different dynamic windows *W* as indicated in the code on top. (**A**) With 256 seismic events, (**B**) with 512 seismic events, (**C**) with 1024 seismic events and (**D**) with 2048 seismic events. Stars give the time of the most important earthquakes of the series whose magnitude is given in the inset.

**Figure 6 entropy-25-01417-f006:**
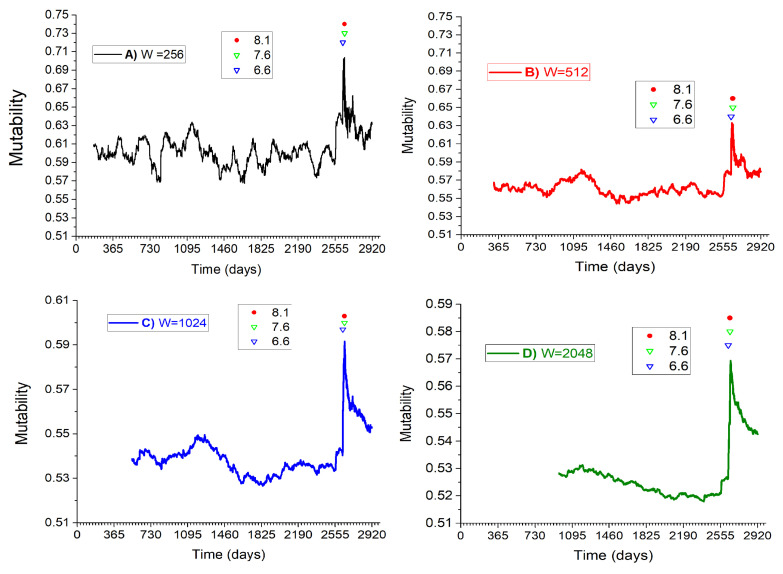
Mutability on magnitude sequence in real time using four different dynamic windows W as indicated in the insets. Construction is similar to previous figure but this time we choose to report the time in days, grouped in years. Stars report the three major earthquakes as given in the inset.

**Figure 7 entropy-25-01417-f007:**
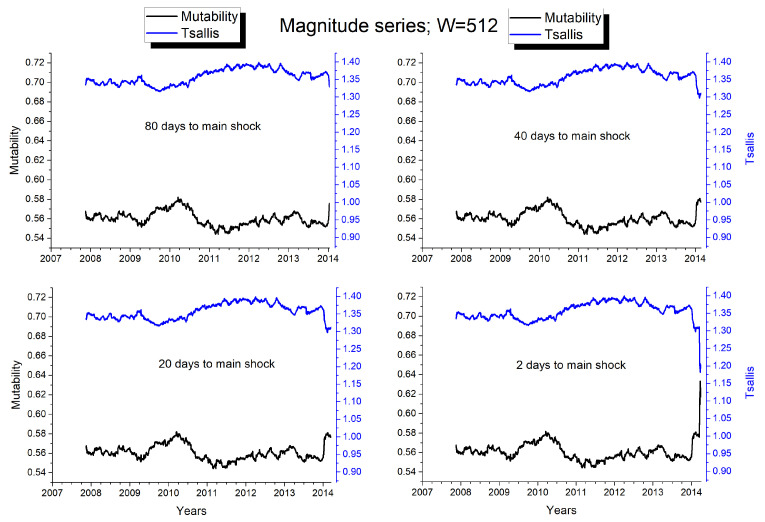
Approximation to the main shock by a dynamical window of W = 512 events. The data is the same in the four plots but time is stopped at 80, 40, 20 or 2 days before the strongest $M_{w}8.1$ quake. Eventually, a video could be a more appropriate way to represent this evolution, but the most relevant information is obtained from these four pictures. The black curve represents mutability and the blue curve Tsallis entropy.

**Figure 8 entropy-25-01417-f008:**
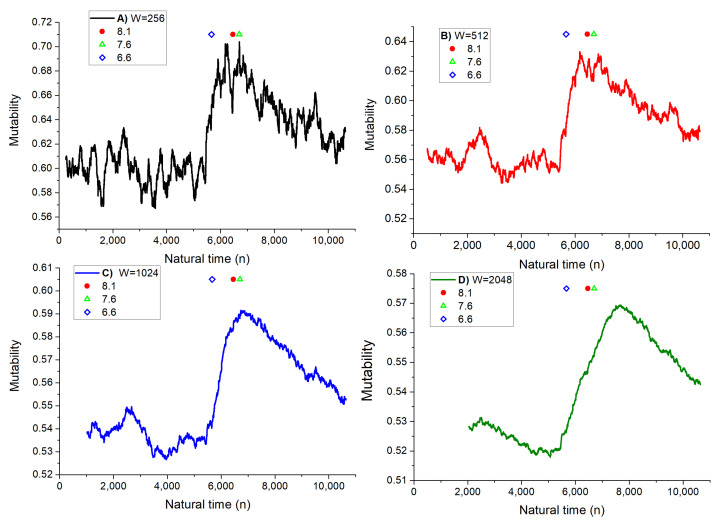
Mutability on magnitude sequence in natural time using four different dynamic windows W as indicated in the insets. Despite that the data is the same as in Figure 6, the texture of the curve looks different. In particular, the stars denoting the main seisms now open up.

**Table 1 entropy-25-01417-t001:** Illustration of the generation of the compressed files. The first column is just an enumeration of lines. The second, third and fourth (sixth, seventh and eighth) columns refer to the rules applied to file “Before” (“After”), giving the magnitude M, the relative coordinates to construct the map and the corresponding frequencies of the magnitudes of that row. Details are given in the text.

	Before			After		
n	**M**	MapB	$f_{B}$	**M**	MapA	$f_{A}$
1	**2.4**	0 18 10 7 4	5	**6.6**	0	1
2	**4.0**	1 2 44,2	4	**4.5**	1	1
3	**3.8**	2 34	2	**4.8**	2 4 5 10	4
4	**2.2**	4 4 14 11,2 8	6	**4.1**	3 26	2
5	**4.6**	5	1	**3.1**	4 6 3 3 22,2 9	7
6	**3.5**	6 11 2	3	**5.2**	5	1
7	**2.9**	7 22 20	3	**4.2**	7	1
8	**3.0**	9	1	**3.8**	8 28	2
9	**4.1**	10 27	2	**3.5**	9 10 4	3
10	**3.1**	11,2	2	**3.7**	12 32 4	3
11	**4.3**	13,2	2	**3.6**	14	1
12	**3.6**	15	1	**3.3**	15 18 10	3
13	**5.5**	16	1	**4.0**	17 14	2
14	**2.7**	20	1	**2.7**	18	1
15	**2.5**	21 2 2 15 2 3	6	**3.2**	20 8	2
16	**2.3**	24 8 12	3	**2.8**	22	1
17	**2.6**	26 4	2	**2.4**	24	1
18	**3.3**	27	1	**4.7**	25 12 4 4	4
19	**2.8**	31 7 5	3	**3.0**	26 20	2
20	**3.4**	46	1	**2.9**	27	1
21				**4.3**	30 4	2
22				**5.1**	32	1
23				**3.4**	35 7 7	3
24				**2.6**	40	1

## Data Availability

The data set used in this article is free and can be downloaded from https://www.ipoc-network.org/welcome-to-ipoc/.

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
