# Peer review of "Tsallis Entropy and Mutability to Characterize Seismic Sequences: The Case of 2007–2014 Northern Chile Earthquakes"

_entropy, 2023, doi:10.3390/e25101417_

Round 1

Reviewer 1 Report

 The paper discusses the estimation of two statistical quantities, Tsallis entropy and mutability (dynamical entropy), to identify indicators for assessing seismic risk in the Iquique region of Northern Chile. The data analyzed covers the period from 2007 to 2014 and focuses on seismic activity within a specific geographic area and depth range. The study identifies two potential forecasting indicators:

i. Tsallis entropy increases and mutability decreases about two years before a major MW 8.1 earthquake, indicating energy accumulation.

ii. Tsallis entropy decreases and mutability increases a few weeks before the MW 8.1 earthquake, suggesting energy relaxation.

The paper discusses the implications of these findings and their relevance for future seismic risk studies.

The paper is well written and organized. I think that the paper can be published after some things are considered, improving the overall quality of the paper.

1. Title: I would add also the information regarding the case study. For instance, I would suggest: “Tsallis Entropy and mutability to characterize seismic sequences: the case of 2007-2014 Northern Chile earthquakes”

2. Figure 2. I would suggest to draw a vertical line in correspondence of the threshold magnitude Mo (by the way, please use in line 134 also the technical term “magnitude of completeness”, as usually said in statistical seismology)

3. A 2D earthquake long-lat distribution in space is missing (Figure 3 shows only longitude-depth distribution).

4. Line 148. Please correct “cause” with “caused”.

5. Where are “eq. A1” (Line 191), “eq. A4”  (Line 202) and “eq. A9” (Line 201) ? I did not find any Appendix with equations.

6. It is not fully clear the spatial and temporal intervals where the S and b are estimated (I presume it is the complete space domain defined in line 28 and repeated in line 132), but it is not explicitly said; and W seems the time interval, but why not considering the number of earthquakes as usually applied in statistical seismology, since is that number that is important and not the interval of time?).

7. Line 202. What is “num”? just an abbreviation for “numerator”? Please write it for extension.

8. Figure 6 has slightly different x units than those in Figures 5 and 7. Why not using the same units, improving the possible comparison of both entropy and mutability results?

9. Figure 5. The Tsallis entropy decreases significantly at the time of the 2014 mainshock, meaning as you state a reduced number of microstates. This is counter-intuitive because we expect that at the time of the earthquake all microstates (e.g. relative a the fracture lengths) are activated, including the largest one that produce the largest rupture. Could you better explain it?

10. Some recent literature on Tsallis Entropy (not coming from the same authors of this paper) is missing. For instance, Skordas et al. 2020 and Sigalotti et al. 2023

10. The presumed improvement of using natural time with respect to conventional time is not straightforward. Could you please better explain the utility of using natural time?

References

Sigalotti, L.D.G.; Ramírez-Rojas, A.; Vargas, C.A. Tsallis q-Statistics in Seismology. Entropy 202325, 408. https://doi.org/10.3390/e25030408

Skordas, E.S., Sarlis, N.V. & Varotsos, P.A. Precursory variations of Tsallis non-extensive statistical mechanics entropic index associated with the M9 Tohoku earthquake in 2011. Eur. Phys. J. Spec. Top. 229, 851–859 (2020). https://doi.org/10.1140/epjst/e2020-900218-x

Author Response

We have responded to the review in the attached pdf file. 

We appreciate the thorough review of the unknown referee.

Reviewer 2 Report

The paper “Tsallis entropy and mutability to characterize seismic sequences” uses two statistical tools to study seismic data before (and after) the large seismic activity near Iquique in March 2014. The statistical tools are the Tsallis entropy and the mutability or dynamical entropy. The choice of the data, that means, the depth of the epicenter and the area extension are well described. Both statistical tools reveals a singular behavior close to the great seismic event. The methodology is tested for several time-series windows. Also, the methodology is tested for several days before the event to certify the methodology and give some hint about the physics behind the phenomenon. Finally, the methodology is tested for time after the seismic event. The results are quite promising, of course (the authors are aware of that) the methodology have to be tested to other major earthquakes.

My opinion is that the paper should be accepted for publication. In the following I point out some minor points that could be improved in the manuscript.

1) The second step about finding best Mo2.2 is not well described in line 136.

2) The authors make clear that they are interested in a study centered seismic events with an epicenter situated until 70km depth. The text does not describes properly the reason the most important events to the study are in this region. Why the large peak of the bi-modal distribution of figure 3B was discarded?

3) In line 181 it is written Eq. A1 instead of Eq. 1

4) Line 358 contains Figs. 5B), 5C) etc while line 386 shows Figs. 6a), 6b) etc.

5) Figure fonts along the manuscript changes a lot. Figure 4 is huge.

Author Response

We have the attached a new version of the article in the pdf file.

We appreciate the good comments in your review.

The paper “Tsallis entropy and mutability to characterize seismic sequences” uses two statistical tools to study seismic data before (and after) the large seismic activity near Iquique in March 2014. The statistical tools are the Tsallis entropy and the mutability or dynamical entropy. The choice of the data, that means, the depth of the epicenter and the area extension are well described. Both statistical tools reveals a singular behavior close to the great seismic event. The methodology is tested for several time-series windows. Also, the methodology is tested for several days before the event to certify the methodology and give some hint about the physics behind the phenomenon. Finally, the methodology is tested for time after the seismic event. The results are quite promising, of course (the authors are aware of that) the methodology have to be tested to other major earthquakes.

We appreciate the consideration to publish our article and your comments.

My opinion is that the paper should be accepted for publication. In the following I point out some minor points that could be improved in the manuscript.

  • The second step about finding best Mo2.2 is not well described in line 136.

Thank you very much for this comment, we have added a brief text to explain better how we find the best .

  • The authors make clear that they are interested in a study centered seismic events with an epicenter situated until 70km depth. The text does not describes properly the reason the most important events to the study are in this region. Why the large peak of the bi-modal distribution of figure 3B was discarded?

Thank you for your question. We have added a brief text to explain this decision. We have a previous work in which we have analyzed the influence of the depth in the analysis of entropy in the same region, so in this work we have decide to focus our analysis in the region close to the hypocenter of the main earthquake. 

  • In line 181 it is written Eq. A1 instead of Eq. 1

Thank you for this comment, we have fixed this error.

  • Line 358 contains Figs. 5B), 5C) etc while line 386 shows Figs. 6a), 6b) etc.

Thank you for this comment, we have fixed this error.

  • Figure fonts along the manuscript changes a lot. Figure 4 is huge.

Thank you for this comment, we have revised the text to use the same fonts. We consider that the map shows in Fig. 4 has an adequate size.

Round 2

Reviewer 1 Report

I am satisfied by the reply and the corresponding revision of the manuscript, except for the point #6 regarding W. As said in the reply, you confirm that this is the number of earthquakes used in subsequent analyses, but in the manuscript it is still written (line 2010) "the time interval W". Please specify differently. Thanks

Author Response

Thank you very much for your sharp review, that line has already been modified.

Reviewer 2 Report

No comments

Author Response

Thank you very much for your review.